# Characterization of the Heat Shock Transcription Factor Family in *Medicago sativa* L. and Its Potential Roles in Response to Abiotic Stresses

**DOI:** 10.3390/ijms241612683

**Published:** 2023-08-11

**Authors:** Hao Liu, Xianyang Li, Yunfei Zi, Guoqing Zhao, Lihua Zhu, Ling Hong, Mingna Li, Shiqing Wang, Ruicai Long, Junmei Kang, Qingchuan Yang, Lin Chen

**Affiliations:** 1Institute of Animal Science, Chinese Academy of Agricultural Sciences, Beijing 100193, China; lhaojy@163.com (H.L.); xianyangli2022@163.com (X.L.); limingna@caas.cn (M.L.); dragongodsgod@163.com (R.L.); kangjmei@126.com (J.K.); qchyang66@163.com (Q.Y.); 2College of Grassland Science, Qingdao Agricultural University, Qingdao 266109, China; 3Institute of Forage Crop Science, Ordos Academy of Agricultural and Animal Husbandry Sciences, Ordos 017000, China; ziyunfei8@163.com (Y.Z.); ordoszhaoguoging@163.com (G.Z.); zhulihua0770@163.com (L.Z.); hl719587523@163.com (L.H.); wangshiqing2888@163.com (S.W.)

**Keywords:** alfalfa, *MsHSF*, genome-wide, gene family, abiotic stress

## Abstract

Heat shock transcription factors (HSFs) are important regulatory factors in plant stress responses to various biotic and abiotic stresses and play important roles in growth and development. The *HSF* gene family has been systematically identified and analyzed in many plants but it is not in the tetraploid alfalfa genome. We detected 104 *HSF* genes (*MsHSF*s) in the tetraploid alfalfa genome (“Xinjiangdaye” reference genome) and classified them into three subgroups: 68 in HSFA, 35 in HSFB and 1 in HSFC subgroups. Basic bioinformatics analysis, including genome location, protein sequence length, protein molecular weight and conserved motif identification, was conducted. Gene expression analysis revealed tissue-specific expression for 13 *MsHSF*s and tissue-wide expression for 28 *MsHSF*s. Based on transcriptomic data analysis, 21, 11 and 27 *MsHSF*s responded to drought stress, cold stress and salt stress, respectively, with seven responding to all three. According to RT–PCR, *MsHSF27*/*33* expression gradually increased with cold, salt and drought stress condition duration; *MsHSF6* expression increased over time under salt and drought stress conditions but decreased under cold stress. Our results provide key information for further functional analysis of *MsHSF*s and for genetic improvement of stress resistance in alfalfa.

## 1. Introduction

Plants are vulnerable to various abiotic stresses during their growth and development, such as high-temperature stress, salt stress, alkali stress and cold stress [1,2]. These abiotic stresses seriously threaten the normal growth and development of plants and even cause death [3]. During evolution, plants have developed various biological mechanisms to address these abiotic stresses [4]. Transcription factors play an important role in the plant response to abiotic stresses [5]. As a family of transcription factors widely present in plants, heat shock transcription factors (HSFs) regulate expression of downstream genes through specific *cis*-regulatory elements, enhancing the ability of plants to cope with different abiotic stresses [6].

Most *HSF* members contain five conserved domains: a DNA-binding domain (DBD) located at the N-terminus, an oligomerization domain (OD or HR-A/B), a nuclear localization signal (NLS), a nuclear export signal (NES) and an activation domain at the C-terminus (CTAD) [7]. Among the five domains, the DBD and OD are the most conserved [8]. The DBD domain specifically recognizes and binds to the conserved motifs of heat shock elements in target genes (5′-AGAAnnTTCT-3′), regulating the expression of downstream stress resistance genes [9]. The HR-A/B domain usually has a coiled-coil structure and is linked to the DBD domain through a flexible connector with variable length [10]. The *HSF* gene family is divided into three subgroups according to the length of the DBD domain from the OD domain and the number of amino acid residues between HR-A and HR-B: A, B and C [11]. The NLS domain is usually composed of basic amino acids, which guide the transport of HSF proteins from the cytoplasm to the nucleus [12]; conversely, the NES domain is usually rich in leucine, which promotes HSF protein export from the nucleus to the cytoplasm [13]. The CTAD domain is the least conserved among the five domains and typically contains the AHA motif, which is composed of large hydrophobic, aromatic and acidic amino acid residues [14]. In addition, the AHA motif is only found in members of the A subgroup and does not exist in the B and C subgroups [15].

The first *HSF* gene in plants was discovered in tomatoes in 1990 [16], and since then, an increasing number of *HSF* genes have been cloned with the continuous completion of reference genomes for different plants. Many studies have shown that *HSF* genes are involved in various processes of plant growth and development. In *A. thaliana*, *AtHSFA9*, which can be activated by *ABI3,* regulates the expression of downstream genes to modulate the process of embryonic development and seed maturation [17]. The 14 *HSF* members found in citrus fruits are all involved in the development and ripening process of fruit, among which *CrHsfB2a* and *CrHsfB5* regulate the citrate content [18]. In addition, *HSF* family members have been confirmed to be widely involved in the response to various abiotic stresses, such as high-temperature stress, drought stress and salt stress. In Arabidopsis, overexpression of the *HsfA2* gene can significantly improve the survival rate of transgenic lines under high temperatures, that is, enhancing their heat tolerance [19]. The *AtHsfA1* gene can regulate the synthesis of downstream heat shock proteins under high temperatures, enabling plants to cope with high-temperature stress [20]. Overexpression of the *SlHsfA3* gene in tomatoes also increases the ability of plants to withstand high-temperature stress [21]. Drought stress, salt stress and cold stress can also induce the gene expression of *HSFs*. In carrots, three *HSF* genes are upregulated under salt stress, and 33 *HSF* genes are downregulated under drought stress [22]. *OsHsfB2b* negatively regulates salt tolerance in rice [23], and overexpression of *AtHsfA1b* improves yield and the harvest index under drought stress [24]. These results indicate that members of the *HSF* gene family not only participate in the normal growth and development of plants but also play an important role in the processes by which plants respond to various abiotic stresses.

To date, 21, 25, 30, 41, 38 and 60 *HSF* members have been found in Arabidopsis [11], rice [25], maize [26], bamboo [27], soybean [28] and *Brassica juncea* [29], respectively. Alfalfa is one of the most important leguminous forage crops in the world, with rich nutritional value and is known as the “king of forage” [30]. At present, three reference genomes of alfalfa have been assembled. The reference genome “Zhongmu No.1” is a haploid genome with a genome size of 816 Mb [31]; reference genomes “Xinjiangdaye” and “Zhongmu No.4” are autotetraploid genomes with genome sizes of 3.15 Gb and 2.74 Gb [32,33], respectively. A previous study showed that there are 16 *MsHSF* members in the “Zhongmu No.1” reference genome, which is a haploid genome [34]. However, *HSF* gene family members remain unidentified in the tetraploid alfalfa reference genome, so it can be considered that more *HSF* genes can be identified in the tetraploid alfalfa reference genome. In this study, the *MsHSF* members were identified in the “Xinjiangdaye” autotetraploid genome. The genome position, gene structure, conserved motifs and *cis*-acting elements in the promoters of these *MsHSF* members were determined. The evolutionary relationship and gene replication events of these *MsHSF* members between alfalfa and *Glycine max*, *M. truncatula* and *A. thaliana* were comprehensively examined. Moreover, in order to determine their potential roles and the response level of each *HSF* gene to different stresses, expression patterns of these *MsHSF* members in six different tissues of alfalfa were analyzed in depth and their dynamic expression changes under drought stress, salt stress and cold stress were analyzed to preliminarily clarify the functions of different *MsHSF* genes in response to abiotic stresses in alfalfa. Our results provide valuable information for further clarifying the molecular regulatory mechanisms of *MsHSF* genes involved in various abiotic stresses in alfalfa in the future.

## 2. Results

### 2.1. MsHSF Gene Identification and Characterization in the Alfalfa Autotetraploid Genome

To identify *MsHSF* gene members in the alfalfa autotetraploid genome “Xinjangdaye”, Hidden Markov model (HMM) analysis and domain analysis were conducted in this study. A total of 104 *MsHSF* genes were found in the “Xinjiangdaye” reference genome. The gene ID, genomic position, length of CD sequence, length of proteins, molecular weight (MW), isoelectric point (pI) and subcellular location of these 104 *MsHSF* genes are shown in Table 1 and Appendix A and Figure 1.

Among the 104 *MsHSF* members, *MsHSF23* has the shortest CD length of 369 bp; *MsHSF16* has the longest CD length of 4506 bp. The protein MWs of these *MsHSF* members range from 14.55 kDa (*MsHSF23*) to 171.49 kDa (*MsHSF16*), and their pI values range from 4.51 (*MsHSF31*) to 8.8 (*MsHSF93*). Based on the results of subcellular location prediction, 101 *MsHSF* members are predicted to localize to the nucleus and *MsHSF23* to the cytosol; two *MsHSF* members (*MsHSF95* and *MsHSF98*) are predicted to localize to the chloroplast.

As shown in Figure 1, the 101 *MsHSF* members (*MsHSF1*–*MsHSF101*) are unevenly distributed on the 32 chromosomes of the “Xinjiangdaye” reference genome, but three *MsHSF* members (*MsHSF102*–*MsHSF104*) are not located on chromosomes. Six *MsHSF* genes are distributed on chr4.4 and chr6.3, the largest. Five *MsHSF* genes are distributed on chr4.1, chr4.3 and chr6.1. Four *MsHSF* genes are located on chr1.1, chr1.3, chr1.4, chr2.3, chr5.2, chr5.3, chr6.2 and chr6.4. Three *MsHSF* genes are distributed on chr1.2, chr2.2, chr4.2, chr5.1, chr5.4 and chr8.4. Only one *MsHSF* gene is located on chr3.1, chr7.2 and chr8.3. Finally, the 104 *MsHSF* genes were renamed based on their position in the reference genome.

### 2.2. Phylogenetic Analysis of HSF Genes in Alfalfa

To understand the classification and evolutionary relationships of *MsHSF* genes in alfalfa, 126 HSF protein sequences, including 104 from alfalfa and 22 from the model plant *Arabidopsis thaliana* [25], were used to construct a phylogenetic tree (Figure 2). Based on the clustering results, the 104 *MsHSF* members in alfalfa can be divided into three major subgroups: HSF-A, HSF-B and HSF-C. There are 68 *MsHSF* members belonging to the A subgroup. The A subgroup was further divided into four subclusters according to the phylogenetic relationship, defined as A1, A2, A3 and A4. Thirty-five *MsHSF* members belong to the B subgroup, with only one *MsHSF* member (*MsHSF72*) belonging to the C subgroup. Proteins in the same class usually have similar biological functions, providing valuable information for predicting the biological function of *MsHSF* members in the future.

### 2.3. Gene Structure and Conserved Motif Analysis of MsHSF Genes

As shown in Figure 3, all 104 *MsHSF* members in alfalfa contain at least one intron. *MsHSF16*, which belongs to subgroup A4, contains the largest number of introns, up to 14. Introns are involved in regulating variable splicing and expression of genes. In addition, we found the length of the introns of 104 *MsHSF* members to be diverse.

To identify conserved motifs among the *MsHSF*s, the MEME tool was used to conduct motif analysis. A total of 10 conserved motifs (Motif1–Motif10) were detected among the 104 *MsHSF* members. The 68 *MsHSF* members belonging to the A subgroup mainly contain Motif1, Motif2, Motif3, Motif4, Motif5, Motif6, Motif7, Motif9 and Motif10 and the 35 *MsHSF* members belonging to the B subgroup mainly Motif1, Motif2, Motif3, Motif6, Motif8, Motif9 and Motif10. *MsHSF72*, which belongs to the C subgroup, mainly contains Motif1, Motif2, Motif3, Motif4 and Motif9. Among the 10 conserved motifs, Motif1, Motif2 and Motif3 are present in the protein sequences of most *MsHSF* members. This result indicates that these three motifs (Motif1, Motif2 and Motif3) may comprise the conserved DBD domain and can be used as a criterion to determine whether a gene is a member of the *HSF* gene family. In addition, we found that some motifs only exist in specific subgroups and *MsHSF* members. For example, Motif8 was only found in the B subgroup of the *MsHSF* gene family, Motif5 only in the A subgroup and Motif7 only in the A4 subgroup. These results suggest that the *MsHSF* genes in the same subgroup have identical structures and conserved motifs and that the diversity of motifs led to the diversity of biological functions among *MsHSF* members.

### 2.4. Gene Duplication Events and Synteny Analysis among MsHSF Genes

To better understand potential gene duplication events among the *MsHSF* genes, tandem duplication and segmental duplication analyses were conducted in this study. As shown in Figure 4, 8 tandem duplication events involving 24 *MsHSF* members were found. For example, *MsHSF21*/*MsHSF22*, a tandem duplication event located at chr2.3, and *MsHSF77*/*78*/*79*/*80*/*81*, another tandem duplication event, involve five *MsHSF* genes located at chr6.3 (Appendix A). In addition, a total of 172 segmental duplication events involving 86 different *MsHSF* members were detected (Appendix A). For example, *MsHSF1*/*MsHSF8*/*MsHSF12* are located on three different chromosomes, chr1.1, chr1.3 and chr1.4, respectively. These results indicate that duplication events have occurred widely among *MsHSF* genes, and that segmental duplication may be the evolutionary driving force of the *MsHSF* gene family in alfalfa.

Next, to clarify potential evolutionary events of the *HSF* gene family in various crops, collinearity maps of alfalfa with *G. max*, *A. thaliana* and *M. truncatula* were constructed. As illustrated in Figure 5, 55, 83 and 75 *MsHSF* genes show collinearity with *A. thaliana*, *M. truncatula* and *G. max*, respectively. Among these genes, there are 80 collinear gene pairs in *A. thaliana*, 118 in *M. truncatula* and 255 in *G. max*. The number of collinear genes between alfalfa and the two legumes (*G. max* and *M. truncatula*) is significantly greater than that between alfalfa and *A. thaliana*, suggesting that the *MsHSF* gene family is relatively conserved among legume plants.

### 2.5. Analysis of Cis-Elements in the Promoter Regions of MsHSF Genes

To further clarify the biological functions of *MsHSF* genes in alfalfa, *cis*-elements of promoter regions located approximately 2 kb upstream of the start codon (ATG) of the *MsHSF* genes were analyzed. As depicted in Figure 6, a total of 11 different *cis*-elements were found. In detail, 74.0% of the *MsHSF* members contain an abscisic acid responsiveness element (ABRE), 64.4% MeJA responsiveness elements (TGACG element and CGTCA element), 59.6% auxin responsiveness elements (AuxRR element, TGA-box element and TGA element), 48.1% GA responsiveness elements (GARE, P-box and TATC box), 37.5% the salicylic acid responsiveness element and 33.7% the zein metabolism regulation element. In addition, we found that some *MsHSF* genes contain specific *cis*-elements in their promoter regions. For example, there are 15 ABREs in the promoter region of *MsHSF33*; only one P-box element was found in the promoter region of *MsHSF30*, and only one TCA element was found in the promoter region of *MsHSF43*. These results indicate that expression of these *MsHSF* genes is likely induced by different hormones and stimuli.

### 2.6. Expression Patterns of MsHSF Genes in M. sativa Tissues

To clarify expression patterns of the *MsHSF* genes in different tissues in alfalfa, transcriptome data for six different alfalfa tissues (leaves, elongated stems, roots, preelongated stems, nodules and flowers) were obtained from a public database (Appendix A). The results showed that 65 *MsHSF* genes were expressed in one or more of the six investigated tissues; the other 39 *MsHSF* genes showed no expression in these tissues, but they might be expressed in different tissues or under specific stress conditions. Overall, expression patterns of the 65 expressed *MsHSF* genes varied in different tissues. As shown in Figure 7, 13 *MsHSF* genes were expressed in only one specific tissue, indicating tissue specificity. For example, *MsHSF29* and *MsHSF30* were only expressed in flowers, *MsHSF4* and *MsHSF96* only in leaves and *MsHSF42*/*70*/*90* only in roots. We also found that 8 *MsHSF* genes were expressed in two different tissues, 7 *MsHSF* genes in three different tissues, 4 *MsHSF* genes in four different tissues, 5 *MsHSF* genes in five different tissues and 28 *MsHSF* genes in six different tissues. Furthermore, the expression abundance of these *MsHSF* genes varied significantly among different tissues. For example, *MsHSF100* was expressed in both roots and flowers, but its expression abundance in roots was significantly higher than that in flowers. *MsHSF6*/*27*/*33* were expressed in six different tissues, but *MsHSF6* was mainly expressed in leaves; *MsHSF27*/*33* was mainly expressed in flowers. These results indicate that these *MsHSF* genes have different functions during normal growth and development.

### 2.7. Expression Analysis for MsHSF Genes under Different Abiotic Stresses

To clarify differential expression levels of the *MsHSF* genes under different abiotic stresses (drought, cold and salt), transcriptomic data of alfalfa seedlings under drought, cold and salt stress were obtained from a public database and analyzed (Appendix A). As presented in Figure 8A–C, 21, 27 and 11 *MsHSF* genes responded to drought stress, salt stress and cold stress, respectively, with some responding to only one abiotic stress. For example, *MsHSF31*/*38* only responded to drought stress, *MsHSF18*/*43*/*50*/*62*/*101* only responded to salt stress and *MsHSF31*/*38* only responded to cold stress in alfalfa. However, other *MsHSF* genes responded to two or three different abiotic stresses, 19 *MsHSF* members responded to both drought and salt stress simultaneously, 10 responded to both cold and salt stress simultaneously and 7 responded to both cold and drought stress simultaneously. Surprisingly, seven *MsHSF* genes (*MsHSF6*/*12*/*27*/*33*/*58*/*82*/*86*) were found to simultaneously respond to cold, drought and salt stress.

To verify the results based on transcriptomic data, an RT–PCR experiment was conducted for three selected genes (*MsHSF27*/*33*/*6*). The related primers are shown in Appendix A. The expression abundance of *MsHSF27*/*33* gradually increased over time under drought, cold and salt stress (Figure 9A–C). The expression abundance of *MsHSF6* increased over time under salt and drought stress but decreased over time under cold stress. As these results are similar to the transcriptome expression results, these genes can be used as candidate genes for further study of their functions in response to abiotic stresses.

### 2.8. Identification and Analysis of Genes Coexpressed with MsHSF6/27/33 under Salt Stress

To investigate whether *MsHSF6*/*27*/*33* is involved in salt stress in alfalfa, genes co-expressed with *MsHSF6*/*27*/*33* were identified based on correlation analysis (Appendix A). As shown in Figure 10, 56 genes correlated significantly with *MsHSF6* under salt stress, 52 with *MsHSF27* under salt stress and 42 with *MsHSF33* under salt stress.

Among the 56 genes co-expressed with *MsHSF6* under salt stress, 10 genes correlated negatively and 46 positively. Among the 46 genes correlating positively with *MsHSF6* under salt stress, some are involved in the plant response to salt stress. For example, the correlation coefficient between the expression abundance of *MsHSF6* and MS.gene067783, which encodes an NAC transcription factor, was 0.99. Expression of MS.gene36742, which encodes the auxin-responsive protein IAA26, also correlated significantly positively with *MsHSF6* under salt stress.

Among the 52 genes co-expressed with *MsHSF27* under salt stress, 8 genes correlated negatively and 44 positively. A previous study showed that *HSF* genes can regulate the expression of heat shock proteins [35]. Of the 8 genes correlating negatively with *MsHSF27*, MS.gene89106 encodes a heat shock protein. CBL-interacting serine/threonine-protein can respond to salt stress in many plants [36]. Among the 44 positively correlating genes, expression of MS.gene025407, which encodes a CBL-interacting serine/threonine-protein kinase, correlated significantly positively with *MsHSF27* under salt stress.

Among the 42 genes co-expressed with *MsHSF33* under salt stress, 9 correlated negatively and 33 positively. Of the nine negatively correlating genes with *MsHSF33*, MS.gene03590 encodes an NAC transcription factor and NAC transcription factors regulate the entire process of plant growth and development, including formation of the plant secondary wall and xylem, root growth, fruit ripening and leaf senescence [37]. Expression of MS.gene072903, which encodes the DEAD-box protein, correlated significantly with *MsHSF33* under salt stress. A previous study showed that the DEAD-box protein can respond to drought and salt stress in plants through the ABA pathway [38].

## 3. Discussion

The *HSF* gene family is one of the important transcription factor families in plant growth and development and in response to abiotic stresses. Previous studies have shown that there are significant differences in the number of *HSF* gene family members in different plant species. Twenty-one *AtHSF* members have been found in Arabidopsis [11]. In soybean, a total of 38 *HSF* genes were identified in the reference genome [28]. In addition, 60 *HSF* members were found in *Brassica juncea* [29]. Overall, the number of *HSF* genes identified in the same species varies due to differences in reference genome versions and identification methods. For example, three different studies found 56, 61 and 82 *TaHSF* genes in wheat [39,40,41]. Interestingly, a previous study reported 16 *MsHSF* genes in alfalfa [34]. However, in our study, a total of 104 *MsHSF* members were found. We speculate that there are two main reasons for such a huge difference. The first is that different versions of the alfalfa reference genome were selected. The previous study used the “Zhongmu NO.1” reference genome, which is a haploid genome with a genome size of 816 Mb, while our study used the “Xinjiangdaye” reference genome, which is an autotetraploid genome with a genome size of 3.15 Gb. The second reason may be that different identification strategies and threshold settings in different studies lead to differences in the final number of *HSF* family members in alfalfa.

A gene family is a group of genes that usually originate from the same ancestor, and there may be multiple copies of this ancestor gene. Whole-genome duplication is one of the main driving forces for the expansion of gene family members in plants [42], and segmental duplication and tandem duplication are two main forms of expansion of gene family members in plants [43]. The genome of alfalfa experienced whole-genome duplication during evolution, and many TEs accumulated, which eventually led to the expansion of the alfalfa genome. Previous studies have shown that segmental duplication plays an important role in the expansion of the *HSF* gene family [7]. In a study of the *HSF* gene family in moso bamboo, 27 segmental duplications and 2 tandem duplication events were detected among the 41 *PeHSF* genes [27]. In wheat, 68.8% of *TaHSF* genes have been involved in segmental duplication events [40]. In our study, 172 segmental duplication events involved 86 *MsHSF* genes, with only 8 tandem duplication events involving 24 *MsHSF* members. These results are consistent with those of previous studies of the *HSF* gene family.

*HSF* genes have been proven by many studies to be widely involved in plant growth and development and various abiotic stress processes, including responses to salt stress, drought stress and high-temperature stress. Phylogenetic tree analysis can help predict the biological function of unknown genes through known gene functions. *AtHSFA9* is activated by *ABI3* to regulate the expression of downstream genes, participating in the process of plant seed maturation and embryo development in Arabidopsis [17]. In this study, we found 68 *MsHSF* genes, with *AtHSFA9* belonging to the same subgroup. Among these *MsHSF* genes, *MsHSF94* was significantly expressed in alfalfa flowers, which are the organs for seed development. Therefore, we inferred that *MsHSF94* also participates in the seed maturation process of alfalfa. Overexpression of *AtHsfA1* and *AtHsfA2* from Arabidopsis and *SlHsfA3* from tomato significantly improves heat tolerance [44,45,46]. In addition to heat stress, many *HSF* genes have been proven to be involved in salt and drought stresses. In *Tamarix hispida*, *ThHSFA1* positively regulates salt tolerance by directly activating the expression of *ThWRKY4* [47]. *OsHsfB2b* negatively regulates drought tolerance and salt tolerance in rice [23]. In a study of the *HSF* gene family in carrots, 3 *HSF* genes were upregulated under salt stress and 33 were downregulated under drought stress, which suggests that these *HSF* genes may be involved in the response to salt and drought stresses [22]. In the present study, we found that 21 and 27 *MsHSF* genes responded to drought stress and salt stress, respectively; moreover, 12 *MsHSF* genes (*MsHSF29*/*89*/*46*/*66*/*14*/*59*/*3*/*10*/*76*/*55*/*92*/*91*) responded to both drought and salt stress. Some studies have also found that *HSF* genes respond to cold stress in plants. For example, there are five *VviHsf* genes in wild Chinese grapevine, and six *PvHsf* genes in common bean were found to respond to cold stress [48,49]. We also found that 11 *MsHSF* genes (*MsHSF1*/*6*/*12*/*22*/*27*/*33*/*51*/*58*/*82*/*86*/*104*) responded to cold stress in alfalfa. Taken together, these results indicate that *HSF* genes play an important role in the response to various abiotic stresses in plants.

Alfalfa is an important forage crop worldwide and has a high content of protein and other nutrients. However, alfalfa often encounters a variety of abiotic stresses during growth and development, resulting in a decline in yield and quality. Therefore, cultivating new alfalfa varieties with strong stress resistance is of great significance for ensuring crop production. Recently, with the development of transgenic and gene editing technology, it is possible to cultivate alfalfa varieties with strong abiotic stress resistance. In this study, we identified 104 *MsHSF* members from the tetraploid genome of alfalfa and found that many *MsHSF* members can respond to drought, salt and cold stress. In future research, these *MsHSF* members can be precisely modified by using transgenic and gene editing technologies to cultivate new alfalfa germplasm. Hence, this study provides valuable information for further research on the biological function of *MsHSF* genes and the molecular mechanism of abiotic stress regulation in alfalfa and other plants.

## 4. Materials and Methods

### 4.1. Identification of MsHSF Genes in the Medicago Sativa Genome

The alfalfa genome was obtained from the Alfalfa Genome project (https://fgshare.com/projects/whole_genome_sequencing_and_assembly_of_Medicago_sativa/66380 (accessed on 20 May 2023)) [32]. Arabidopsis protein sequences were obtained from The Arabidopsis Information Resource (TAIR) (https://www.arabidopsis.org/ (accessed on 20 May 2023)), and the *Medicago truncatula* genome was obtained from the website (http://www.medicagogenome.org/ (accessed on 20 May 2023)). Hidden Markov model (HMM) analysis was carried out for the required sequence search, and the Pfam database (https://pfam.xfam.org/ (accessed on 21 May 2023)) was used to obtain the HMM configuration file for *HSF* domains (PF00447) [50]. A total of 104 *MsHSF* genes were identified in the alfalfa genome using BLAST, with a cutoff value of E-value > 1e^−10^. The identified *MsHSF*s were submitted to NCBI Conserved Domain Database (CDD, https://www.ncbi.nlm.nih.gov/cdd (accessed on 22 May 2023)) to check for the existence of conserved structural domains.

### 4.2. Chromosome Location and Gene Information

The chromosome information of *MsHSF*s was visualized using TBtools software (v1.108, Chen, C., GZ, China) [51]. *MsHSF*s were renamed according to the position of the gene on each chromosome. The characteristics of the identified *MsHSF* gene, including CD length, protein length, MW and pI, were studied using the Expasy website (https://web.expasy.org/compute_pi/ (accessed on 22 May 2023)).

### 4.3. Phylogenetic Analysis

The protein sequences used to construct phylogenetic trees were obtained from the UniProt database (https://www.UniProt.org (accessed on 23 May 2023)), and the phylogenetic trees were constructed by using MEGA software (v11.0, Tamura, K., Tokyo, Japan) [52] with the protein sequences of alfalfa and Arabidopsis *HSF* family genes. Clusterx2.0 software was used to compare multiple amino acid sequences of the identified *MsHSF* genes with default parameters. A phylogenetic tree was constructed using the neighbor-joining (NJ) method and 1000 bootstrap replicates were performed. The Poisson correction method was used to calculate evolutionary distance. The grouping of *MsHSF* refers to the method of Guo et al. [25].

### 4.4. Gene Structure, Motif Identification and Conserved Domains

The intron–exon distribution of *MsHSF* genes was obtained by using the GFF file for the alfalfa genome. Conserved amino acid sequences of HSF proteins were analyzed by the online MEME suite (https://meme-suite.org/meme/tools/meme (accessed on 24 May 2023)), and the maximum motif number was set to 10. The NCBI conserved domain database (https://www.ncbi.nlm.nih.gov/cdd/ (accessed on 24 May 2023)) was used to predict conserved domains in *MsHSF*.

### 4.5. Gene Duplication and Synteny Analysis

MCScanX software (http://chibba.pgml.uga.edu/mcscan2/ accessed on 23 May 2023) (Hu, Y., Herndon, VA, USA) [53] was used to determine replication events in *MsHSF* genes and identify collinear regions between them and *HSF* genes in *M. truncatula*, *A. thaliana* and *G. max*. TBtools software was employed to extract information related to gene function and chromosomal location [51].

### 4.6. Identification of Cis-Acting Elements

PlantCARE (http://bioinformatics.psb.ugent.be/webtools/plantcare/html/ (accessed on 22 May 2023)) was used to identify *cis*-acting elements. The upstream 2000 bp sequence was defined as the promoter region for predicting *cis*-acting elements.

### 4.7. Transcriptomic Data Analysis

Transcriptomic data for six alfalfa tissues (leaves, nodules, elongated stems, flowers, preelongated stems and roots) were retrieved from the NCBI database (SRP055547) [54]. Transcriptomic data for the *MsHSF* genes from the alfalfa plants subjected to drought, cold and salt stresses were also obtained from the NCBI database (SRR7160313-SRR7160357 and SRR7091780-SRR7091794) [55]. The obtained clean reads were mapped using TopHat2 [56] to the “Xinjiangdaye” reference genome. Gene expression levels were calculated based on the fragments per kilobase of exon per million mapped fragments (FPKM) value; differentially expressed genes were retrieved using DESeq with the following parameters: padj < 0.05 and |log2FC| ≥ 1 [57]. Data were visualized by TBtools software.

### 4.8. Details of Plant Material and Treatment

Seeds of the “Zhongmu No.1” cultivar of alfalfa were grown at the Institute of Animal Science, Chinese Academy of Agricultural Sciences. Briefly, seeds were first treated for 3 days at 4 °C before germination. Next, the seeds were cultured in a greenhouse under a light/dark (16/8 h) cycle with 70–80% relative humidity and a day/night temperature of 24 °C/20 °C for 2 weeks in hydroponic culture medium. Three stress conditions (salt, cold and drought) were then applied to the cultured plants. For drought conditions, treatment with mannitol (400 mM) was applied to simulate drought stress. After treatment, root tip samples were collected at the following 6 time points: CK at 0 h and M1, M2, M3, M4 and M5 at 1, 3, 6, 12 and 24 h, respectively. For cold treatment, leaves were placed at 4 °C, and the following 5 time points were selected for sampling: 0 h as CK and 2, 6, 24 and 48 h as C1, C2, C3 and C4, respectively. Simulated salt stress involved treatment with NaCl (250 mM), and root tip samples were collected at 7 time points (0 h as CK and 0.5, 1, 3, 6, 12 and 24 h as S1 to S6, respectively). Each stress treatment condition had three replicates, with 5 individual seedlings in each replicate. The samples were stored at −80 °C for subsequent RT–PCR analysis.

### 4.9. Expression Analysis of MsHSF Genes

Total RNA was extracted from all samples in this study using TRIzol reagent according to the manufacturer’s instructions. The corresponding cDNA was obtained using the EasyScript first-strand cDNA Synthesis kit. The primers used in the study were designed using Primer 5.0 software. The RT–PCR experiment was performed using SYBR Premix Ex Taq (Takala, Japan) and a 7500 real-time fluorescent quantitative PCR system (Applied Biosystems, Foster City, CA, USA). Three replicates were designed for each sample, and data were normalized using alfalfa actin gene expression. The relative gene expression level of *MsHSF* genes was calculated by the 2^−ΔΔCT^ [58] method and the results were visualized in TBtools.

### 4.10. Genes Coexpressed with MsHSFs

Genes co-expressed with *MsHSF* were screened based on differentially expressed genes under salt stress in alfalfa, and the standard correlation coefficient was |R| > 0.98.

## 5. Conclusions

In this study, the *HSF* gene family in the “Xinjiangdaye” reference genome of alfalfa, which is an autotetraploid genome, was comprehensively identified and characterized. A total of 104 *MsHSF* genes were found; 101 *MsHSF* genes are unevenly distributed on 32 chromosomes of this genome, and the other 3 *MsHSF* genes were not found on chromosomes. Phylogenetic tree analysis indicates that the 104 *MsHSF* genes can be grouped into classes A, A1, A2, A3, A4, B and C, which is consistent with results for Arabidopsis. Compared with tandem duplication, segment duplication is the main driving force for the expansion of the *MsHSF* gene family in alfalfa. The expression patterns of the 104 *MsHSF* genes in six different tissues in alfalfa revealed that 13 *MsHSF* genes have tissue-specific expression but that 28 *MsHSF* genes are expressed in all six tissues studied. Based on transcriptome data analysis, 21, 27 and 11 *MsHSF* genes respond to drought, salt and cold stresses, respectively. In addition, 10 *MsHSF* genes respond to both cold stress and salt stress, 19 *MsHSF* genes respond to both drought stress and salt stress, 7 *MsHSF* genes respond to both cold stress and drought stress and 7 *MsHSF* genes respond to all three stresses. According to RT–PCR results, *MsHSF27*/*33* expression gradually increased with time under cold, salt and drought stresses, and *MsHSF6* expression increased with time under salt and drought stresses but decreased with time under cold stress. This study provides data to guide further research on how *HSF* gene family members respond to abiotic stress conditions and improve alfalfa quality.

## Figures and Tables

**Figure 1 ijms-24-12683-f001:**
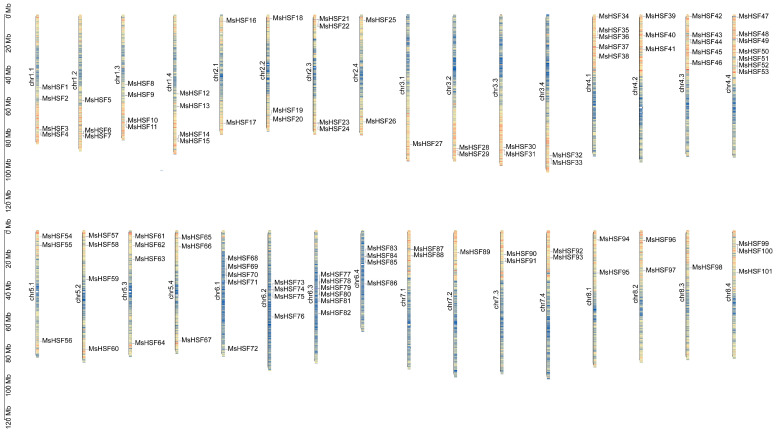
Schematic diagram of the chromosomal distribution of *HSF* genes in *Medicago sativa*.

**Figure 2 ijms-24-12683-f002:**
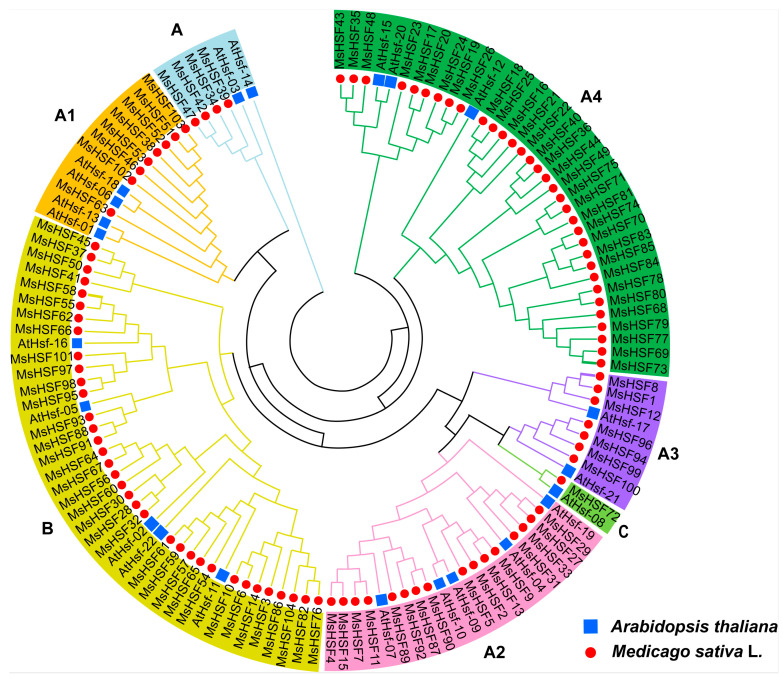
Phylogenetic tree of *HSF* genes in *Medicago sativa* and *Arabidopsis thaliana*.

**Figure 3 ijms-24-12683-f003:**
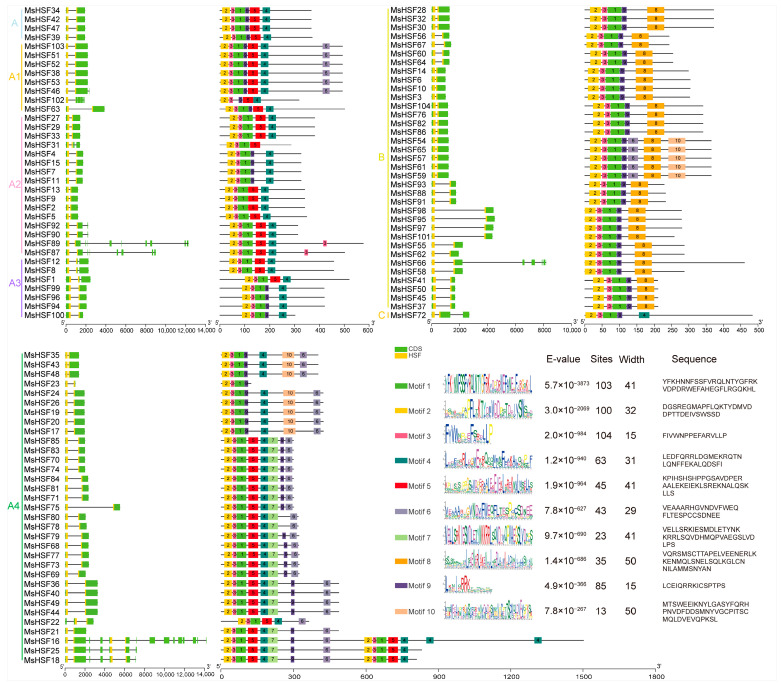
Gene structures and motif compositions of *MsHSF* genes of *Medicago sativa*.

**Figure 4 ijms-24-12683-f004:**
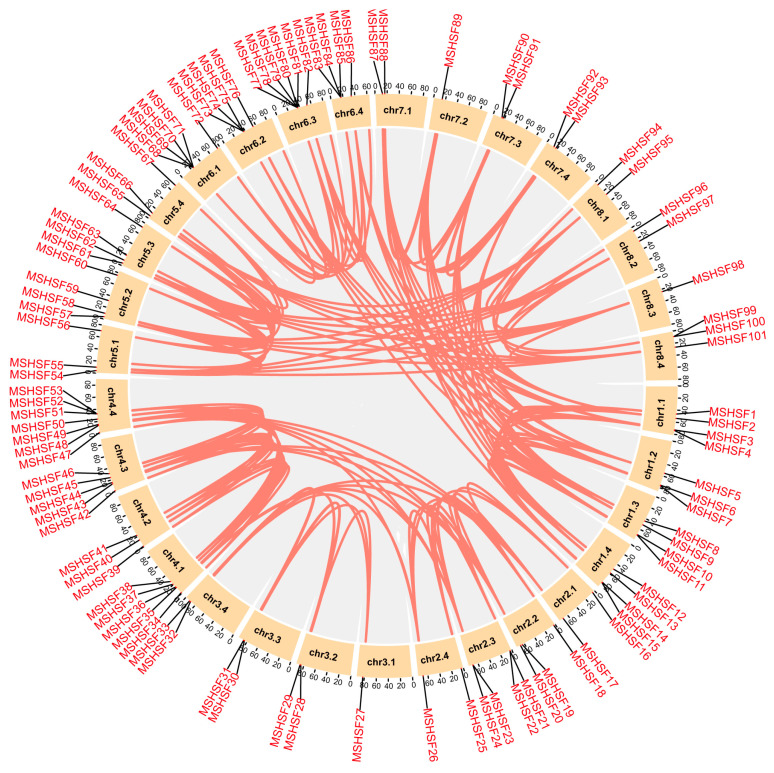
Schematic diagram of syntenic relationships of *MsHSF* genes in *Medicago sativa*. Gray ribbons represent syntenic blocks in the alfalfa genome, and segmental duplication events are marked in red.

**Figure 5 ijms-24-12683-f005:**
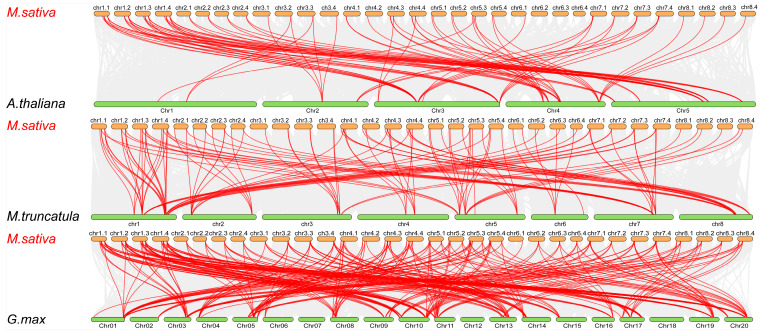
Synteny analysis of the *HSF* genes between *Medicago sativa* and three representative plant species. Gray lines in the background indicate collinear blocks between *M. sativa* and the indicated plant species, whereas the red lines highlight syntenic *HSF* gene pairs.

**Figure 6 ijms-24-12683-f006:**
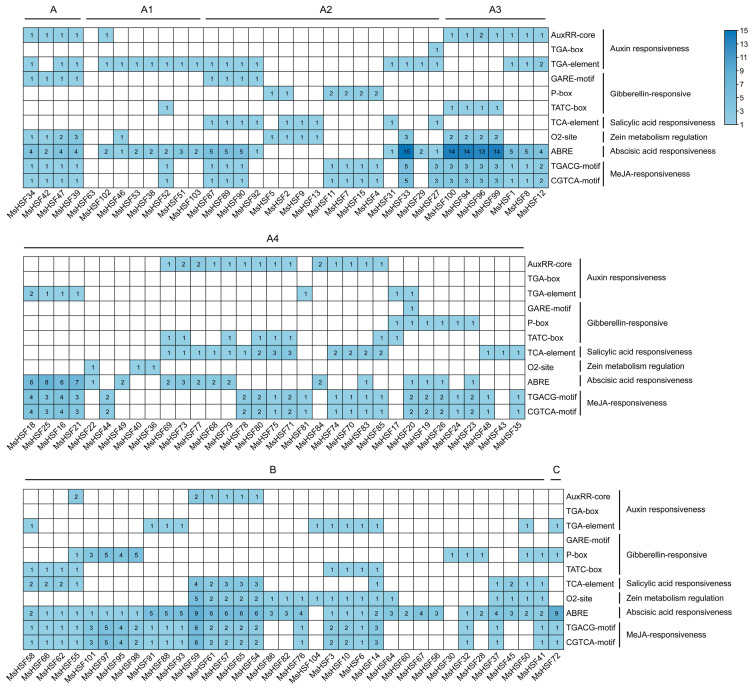
Distribution of *cis*-acting elements related to hormone responses in promoter regions of *MsHSF* genes.

**Figure 7 ijms-24-12683-f007:**
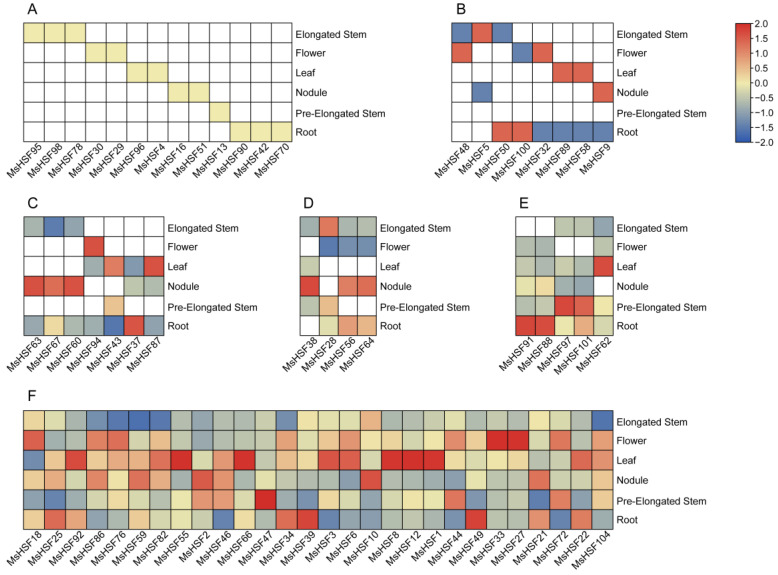
Expression analysis of *MsHSF* genes in different tissues (flowers, leaves, elongated stems, preelongated stems, nodules and roots). (**A**–**F**). *MsHSF* gene expression in 1–6 tissues.

**Figure 8 ijms-24-12683-f008:**
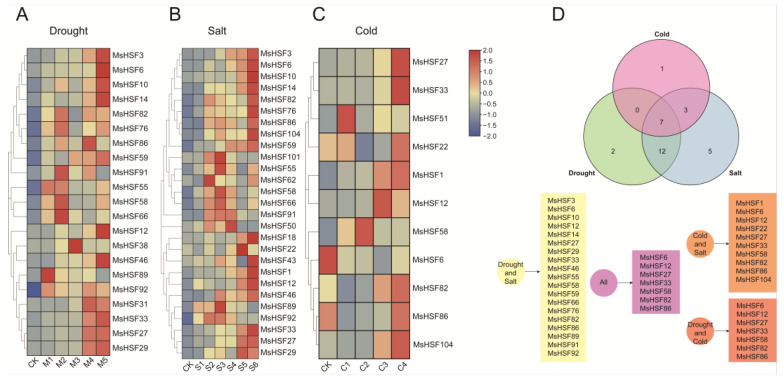
Expression of *MsHSF* genes under drought, salt and cold stress conditions. (**A**). Expression of *MsHSF* genes under drought stress. (**B**). Expression of *MsHSF* genes under salt stress. (**C**). Expression of *MsHSF* genes under cold stress. (**D**). Venn diagram of *MsHSF* genes expressed under the three abiotic stresses.

**Figure 9 ijms-24-12683-f009:**
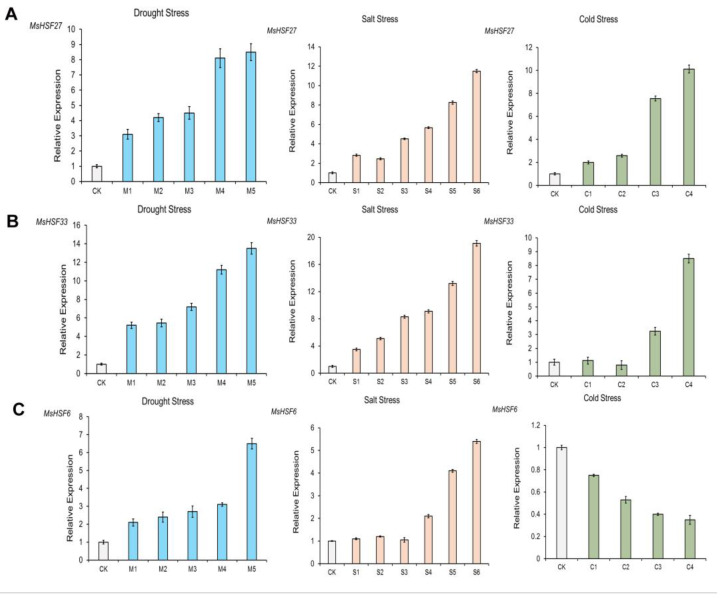
RT–PCR expression of *MsHSF27*/*33*/*6* under drought, salt and cold stress conditions. (**A**). Expression of *MsHSF27*. (**B**). Expression of *MsHSF33*. (**C**). Expression of *MsHSF6*.

**Figure 10 ijms-24-12683-f010:**
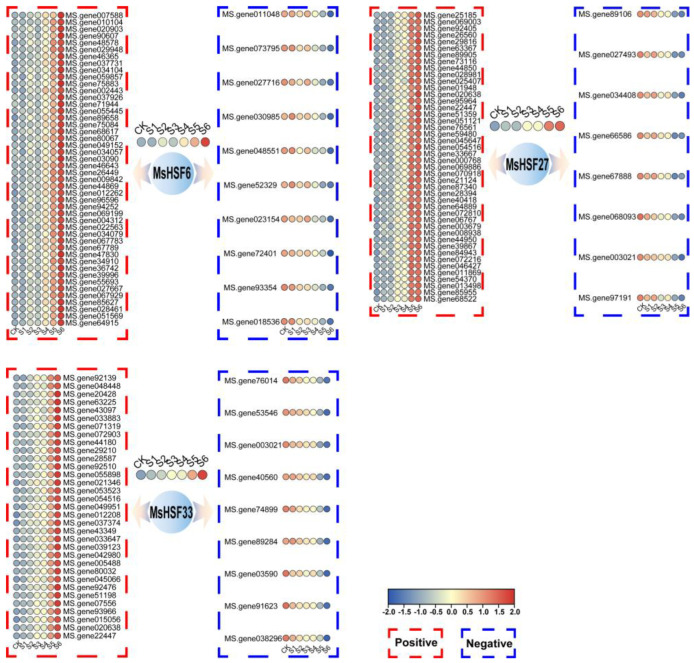
Genes co-expressed with *MsHSF6*, *MsHSF27* and *MsHSF33* under salt stress.

**Table 1 ijms-24-12683-t001:** Details of the *HSF* family genes in *Medicago sativa*.

Gene Name	Gene ID	Chr Location	CDS Length (bp)	Protein Length (aa)	MW (kDa)	pI	Subcellular Location
*MsHSF1*	MS.gene47348	chr1.1:46,491,236–46,493,679	1560	520	59.08	5.5	Nucleus
*MsHSF2*	MS.gene49374	chr1.1:53,676,456–53,677,640	1023	341	40.17	5.44	Nucleus
*MsHSF3*	MS.gene006320	chr1.1:72,806,609–72,807,606	909	303	33.80	7.58	Nucleus
*MsHSF4*	MS.gene37701	chr1.1:73,816,413–73,818,125	978	326	38.08	5.72	Nucleus
*MsHSF5*	MS.gene50549	chr1.2:54,664,413–54,665,621	1047	349	41.23	5.7	Nucleus
*MsHSF6*	MS.gene34469	chr1.2:75,570,174–75,571,157	912	304	33.93	7.56	Nucleus
*MsHSF7*	MS.gene34387	chr1.2:76,550,116–76,551,779	978	326	38.16	5.77	Nucleus
*MsHSF8*	MS.gene034932	chr1.3:44,215,023–44,217,276	1374	458	51.61	4.87	Nucleus
*MsHSF9*	MS.gene036186	chr1.3:51,302,463–51,303,647	1023	341	40.17	5.44	Nucleus
*MsHSF10*	MS.gene49062	chr1.3:69,715,698–69,716,687	912	304	33.85	6.55	Nucleus
*MsHSF11*	MS.gene006403	chr1.3:70,669,752–70,671,419	978	326	38.16	5.77	Nucleus
*MsHSF12*	MS.gene72412	chr1.4:50,359,380–50,361,633	1374	458	51.59	4.87	Nucleus
*MsHSF13*	MS.gene39940	chr1.4:58,394,993–58,396,193	1023	341	40.16	5.44	Nucleus
*MsHSF14*	MS.gene41406	chr1.4:78,626,550–78,627,540	897	299	33.40	8.15	Nucleus
*MsHSF15*	MS.gene070958	chr1.4:79,551,444–79,553,156	978	326	38.17	5.9	Nucleus
*MsHSF16*	MS.gene069887	chr2.1:4,201,900–4,215,604	4506	1502	171.49	7.62	Nucleus
*MsHSF17*	MS.gene002630	chr2.1:68,933,280–68,935,167	1269	423	48.37	4.98	Nucleus
*MsHSF18*	MS.gene36373	chr2.2:2,555,932–2,562,788	2430	810	91.82	5.29	Nucleus
*MsHSF19*	MS.gene001530	chr2.2:63,861,920–63,863,788	1266	422	48.14	4.97	Nucleus
*MsHSF20*	MS.gene01615	chr2.2:66,867,654–66,869,539	1269	423	48.37	4.98	Nucleus
*MsHSF21*	MS.gene76552	chr2.3:3,084,334–3,086,356	1458	486	54.63	5.16	Nucleus
*MsHSF22*	MS.gene76553	chr2.3:3,087,812–3,090,547	1089	363	41.23	4.61	Nucleus
*MsHSF23*	MS.gene004307	chr2.3:68,816,734–68,817,708	369	123	14.55	6.28	Cytosol
*MsHSF24*	MS.gene03101	chr2.3:68,965,280–68,967,152	1266	422	48.19	4.94	Nucleus
*MsHSF25*	MS.gene85167	chr2.4:3,824,061–3,831,001	2493	831	94.39	5.5	Nucleus
*MsHSF26*	MS.gene004306	chr2.4:68,163,087–68,164,955	1266	422	48.14	4.97	Nucleus
*MsHSF27*	MS.gene32806	chr3.1:82,443,497–82,444,921	1143	381	42.84	4.93	Nucleus
*MsHSF28*	MS.gene38358	chr3.2:84,790,567–84,791,844	1116	372	42.09	8.16	Nucleus
*MsHSF29*	MS.gene015008	chr3.2:86,383,022–86,384,435	1143	381	42.83	4.93	Nucleus
*MsHSF30*	MS.gene38707	chr3.3:84,309,749–84,311,026	1116	372	42.08	8.16	Nucleus
*MsHSF31*	MS.gene066508	chr3.3:85,975,236–85,976,635	858	286	31.93	4.51	Nucleus
*MsHSF32*	MS.gene012969	chr3.4:89,680,036–89,681,313	1116	372	42.09	8.16	Nucleus
*MsHSF33*	MS.gene37417	chr3.4:91,525,254–91,526,669	1143	381	42.86	4.82	Nucleus
*MsHSF34*	MS.gene015638	chr4.1:1,466,842–1,468,746	1101	367	41.88	5.07	Nucleus
*MsHSF35*	MS.gene27848	chr4.1:13,157,657–13,158,981	1203	401	45.93	5.33	Nucleus
*MsHSF36*	MS.gene09269	chr4.1:14,521,522–14,524,663	1461	487	54.36	5.06	Nucleus
*MsHSF37*	MS.gene31937	chr4.1:20,930,615–20,932,282	633	211	24.50	5.82	Nucleus
*MsHSF38*	MS.gene006573	chr4.1:26,940,377–26,942,558	1479	493	55.02	5.1	Nucleus
*MsHSF39*	MS.gene62673	chr4.2:1,228,383–1,230,289	1113	371	42.19	5.07	Nucleus
*MsHSF40*	MS.gene08822	chr4.2:13,315,350–13,318,491	1461	487	54.36	5.06	Nucleus
*MsHSF41*	MS.gene39449	chr4.2:22,136,602–22,138,266	630	210	24.41	6.25	Nucleus
*MsHSF42*	MS.gene95836	chr4.3:1,296,902–1,298,804	1101	367	41.90	5.07	Nucleus
*MsHSF43*	MS.gene065968	chr4.3:13,372,525–13,373,849	1203	401	45.93	5.33	Nucleus
*MsHSF44*	MS.gene023483	chr4.3:15,605,046–15,608,184	1461	487	54.36	5.06	Nucleus
*MsHSF45*	MS.gene052478	chr4.3:24,186,653–24,188,321	633	211	24.56	6.02	Nucleus
*MsHSF46*	MS.gene031737	chr4.3:30,991,732–30,994,076	1479	493	55.18	5.01	Nucleus
*MsHSF47*	MS.gene058589	chr4.4:1,483,835–1,485,751	1101	367	41.83	5.02	Nucleus
*MsHSF48*	MS.gene023615	chr4.4:13,251,720–13,253,044	1203	401	45.93	5.33	Nucleus
*MsHSF49*	MS.gene08977	chr4.4:14,900,400–14,903,540	1461	487	54.35	5.1	Nucleus
*MsHSF50*	MS.gene33789	chr4.4:23,836,186–23,837,855	633	211	24.54	6.02	Nucleus
*MsHSF51*	MS.gene065779	chr4.4:32,552,978–32,555,159	1479	493	55.02	5.1	Nucleus
*MsHSF52*	MS.gene065778	chr4.4:32,564,463–32,566,644	1479	493	55.02	5.1	Nucleus
*MsHSF53*	MS.gene006572	chr4.4:32,578,622–32,580,803	1479	493	55.02	5.1	Nucleus
*MsHSF54*	MS.gene015551	chr5.1:4,238,143–4,239,348	1095	365	40.19	5.1	Nucleus
*MsHSF55*	MS.gene015144	chr5.1:9,692,198–9,694,414	861	287	32.19	8.37	Nucleus
*MsHSF56*	MS.gene016952	chr5.1:70,548,465–70,549,718	729	243	28.31	7.08	Nucleus
*MsHSF57*	MS.gene050384	chr5.2:3,806,018–3,807,223	1095	365	40.19	5.1	Nucleus
*MsHSF58*	MS.gene041094	chr5.2:9,587,990–9,590,203	861	287	32.21	6.47	Nucleus
*MsHSF59*	MS.gene050386	chr5.2:31,314,868–31,316,072	1095	365	40.19	5.1	Nucleus
*MsHSF60*	MS.gene47664	chr5.2:75,725,943–75,727,205	762	254	29.59	6.45	Nucleus
*MsHSF61*	MS.gene072808	chr5.3:4,227,992–4,229,197	1095	365	40.19	5.1	Nucleus
*MsHSF62*	MS.gene047640	chr5.3:9,771,172–9,773,103	861	287	32.24	7.55	Nucleus
*MsHSF63*	MS.gene78932	chr5.3:18,647,929–18,651,784	1506	502	55.35	4.73	Nucleus
*MsHSF64*	MS.gene70849	chr5.3:71,945,341–71,946,606	762	254	29.54	6.45	Nucleus
*MsHSF65*	MS.gene019282	chr5.4:4,925,106–4,926,311	1095	365	40.19	5.1	Nucleus
*MsHSF66*	MS.gene010367	chr5.4:10,594,467–10,602,650	1383	461	51.42	5.26	Nucleus
*MsHSF67*	MS.gene038168	chr5.4:70,232,760–70,234,141	729	243	28.26	7.08	Nucleus
*MsHSF68*	MS.gene054313	chr6.1:24,564,039–24,566,289	963	321	37.00	5.47	Nucleus
*MsHSF69*	MS.gene054312	chr6.1:24,572,932–24,574,908	969	323	37.00	5.19	Nucleus
*MsHSF70*	MS.gene054311	chr6.1:24,590,934–24,592,848	897	299	34.29	5.05	Nucleus
*MsHSF71*	MS.gene054310	chr6.1:24,626,677–24,628,933	900	300	34.31	5.14	Nucleus
*MsHSF72*	MS.gene052458	chr6.1:75,626,062–75,628,744	1452	484	54.99	6.14	Nucleus
*MsHSF73*	MS.gene03506	chr6.2:37,838,394–37,840,681	969	323	37.04	5.12	Nucleus
*MsHSF74*	MS.gene03507	chr6.2:37,852,586–37,854,503	897	299	34.40	5.17	Nucleus
*MsHSF75*	MS.gene03509	chr6.2:37,891,892–37,897,203	900	300	34.26	5.37	Nucleus
*MsHSF76*	MS.gene98248	chr6.2:55,114,403–55,115,562	1023	341	37.74	5.64	Nucleus
*MsHSF77*	MS.gene80513	chr6.3:36,909,010–36,911,297	969	323	37.02	5.18	Nucleus
*MsHSF78*	MS.gene80509	chr6.3:36,932,369–36,934,439	957	319	36.77	5.2	Nucleus
*MsHSF79*	MS.gene80508	chr6.3:36,950,527–36,952,807	966	322	37.10	5.3	Nucleus
*MsHSF80*	MS.gene80507	chr6.3:36,994,105–36,996,079	957	319	36.44	5.27	Nucleus
*MsHSF81*	MS.gene80504	chr6.3:37,053,440–37,055,696	900	300	34.31	5.22	Nucleus
*MsHSF82*	MS.gene84114	chr6.3:52,909,236–52,910,395	1023	341	37.78	5.5	Nucleus
*MsHSF83*	MS.gene42038	chr6.4:16,627,031–16,628,940	897	299	34.34	5.31	Nucleus
*MsHSF84*	MS.gene42037	chr6.4:16,648,745–16,650,960	897	299	34.34	4.97	Nucleus
*MsHSF85*	MS.gene42036	chr6.4:16,658,343–16,660,257	897	299	34.32	5.32	Nucleus
*MsHSF86*	MS.gene000756	chr6.4:34,113,335–34,114,494	1023	341	37.75	5.64	Nucleus
*MsHSF87*	MS.gene018149	chr7.1:12,337,762–12,346,750	1506	502	57.70	6.02	Nucleus
*MsHSF88*	MS.gene42537	chr7.1:16,183,243–16,184,979	699	233	26.99	8.2	Nucleus
*MsHSF89*	MS.gene43382	chr7.2:14,288,001–14,300,261	1731	577	66.18	5.84	Nucleus
*MsHSF90*	MS.gene054891	chr7.3:15,394,234–15,396,451	939	313	35.97	6.12	Nucleus
*MsHSF91*	MS.gene017872	chr7.3:17,787,214–17,788,956	699	233	26.99	8.2	Nucleus
*MsHSF92*	MS.gene020288	chr7.4:13,679,001–13,681,206	939	313	35.99	6.43	Nucleus
*MsHSF93*	MS.gene39054	chr7.4:17,608,870–17,610,596	687	229	26.69	8.8	Nucleus
*MsHSF94*	MS.gene012268	chr8.1:6,048,162–6,050,222	1263	421	47.82	5.41	Nucleus
*MsHSF95*	MS.gene011821	chr8.1:27,164,988–27,169,493	837	279	30.93	6.48	Chloroplast
*MsHSF96*	MS.gene051856	chr8.2:6,520,812–6,522,869	1260	420	47.65	5.4	Nucleus
*MsHSF97*	MS.gene56773	chr8.2:25,831,390–25,835,797	840	280	31.02	6.18	Nucleus
*MsHSF98*	MS.gene90514	chr8.3:23,968,421–23,972,840	837	279	30.93	6.48	Chloroplast
*MsHSF99*	MS.gene57572	chr8.4:9,015,415–9,017,472	1257	419	47.51	5.47	Nucleus
*MsHSF100*	MS.gene57604	chr8.4:9,044,302–9,046,006	906	302	34.88	8.68	Nucleus
*MsHSF101*	MS.gene99195	chr8.4:26,635,298–26,639,638	774	258	28.49	5.92	Nucleus
*MsHSF102*	MS.gene065776	33,245:8758–10,604	954	318	35.33	6.09	Nucleus
*MsHSF103*	MS.gene065780	33,246:5248–7429	1479	493	55.02	5.1	Nucleus
*MsHSF104*	MS.gene90989	8272:103,717–104,876	1023	341	37.73	5.5	Nucleus

chr: chromosome; CDS: coding sequence; bp: base pair; aa: amino acid; MW: molecular weight; pI: isoelectric point.

## Data Availability

Not applicable.

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
