# Peer review of "Characterization of the Heat Shock Transcription Factor Family in Medicago sativa L. and Its Potential Roles in Response to Abiotic Stresses"

_ijms, 2023, doi:10.3390/ijms241612683_

Round 1

Reviewer 1 Report

The article "Characterization of the heat shock transcription factor family in Medicago sativa L. and its potential roles in response to abiotic stresses" provides valuable insights and adds new perspectives into alfafa's heat shot transcription factors. The results have high potential on the development of new adapted cultivars for abiotic stresses. 

The paper is technically sound and the methods used by the authors support the results. I don't have major concerns about the paper although I feel that some improvements would benefit the quality of the manuscript and the readers experience.

My minor concerns are:

1) Some figures have really low quality and small letters, being difficult to understand their content, e.g., Fig. 1, Fig. 3, Fig. 5 and Fig. 10.

2) The final paragraph of discussion could draw better a conclusion and including future perspectives.

Author Response

Response to Reviewer 1 Comments

Point 1: Some figures have really low quality and small letters, being difficult to understand their content, e.g., Fig. 1, Fig. 3, Fig. 5 and Fig. 10.

Response 1: Thanks for this advice. We have increased the letters of these figures and changed the picture format to a more clear TIFF format.

Point 2: The final paragraph of discussion could draw better a conclusion and including future perspectives.

Response 2: Thanks for this advice. We have made changes in lines 364-377 as follows:

Alfalfa is an important forage crop worldwide and has a high content of protein and other nutrients. However, alfalfa often encounters a variety of abiotic stresses during growth and development, resulting in a decline in yield and quality. Therefore, cultivating new alfalfa varieties with strong stress resistance is of great significance for ensuring crop production. Recently, with the development of transgenic and gene editing technology, it is possible to cultivate alfalfa varieties with strong abiotic stress resistance. In this study, we identified 104 MsHSF members from the tetraploid genome of alfalfa, and found that many MsHSF members can respond to drought, salt and cold stress. In future research, these MsHSF members can be precisely modified by using transgenic and gene editing technologies to cultivate new alfalfa germplasm. Hence, this study provides valuable information for further research on the biological function of MsHSF genes and the molecular mechanism of abiotic stress regulation in alfalfa and other plants.

Reviewer 2 Report

The article by H. Liu, X. Li, Y. Zi, G. Zhao, L. Zhu, L. Hong, M. Li, S. Wang, R. Long, J. Kang, Q. Yang, and L. Chen entitled «Characterization of the heat shock transcription factor family in Medicago sativa L. and its potential roles in response to abiotic stresses» are devoted to heat shock transcription traits of tetraploid alfalfa. The authors found 104 MsHSF genes.  They showed tissue-specific expression of some gene family. Groups of genes were identified, the expression of which increased during drought, salinity and cold stresses.

There are a few notes to the article:

1) The last paragraph of the introduction is more like an abstract than an introduction. Usually the introduction ends with the authors' assumptions (or working hypothesis or null hypothesis). What did you think when you started this investigation? Did you expect from the literature that more MsHSF genes would be found than in previous studies; that MsHSF genes will be expressed in different tissues; that the same MsHSF genes will respond to different stresses?

2) Alkali stress.  This stress is mentioned twice in the article. First, in the introduction (this is appropriate, since we are talking about abiotic factors). The second time was in the caption to figure 8 (inappropriate, since the article is devoted to only three stress factors, and this one is the fourth one).

Author Response

Response to Reviewer 2 Comments

Point 1: The last paragraph of the introduction is more like an abstract than an introduction. Usually the introduction ends with the authors' assumptions (or working hypothesis or null hypothesis). What did you think when you started this investigation? Did you expect from the literature that more MsHSF genes would be found than in previous studies; that MsHSF genes will be expressed in different tissues; that the same MsHSF genes will respond to different stresses?

Response 1: Thanks for this advice. We have made changes in lines 99-113 as follows:

However, HSF gene family members remain unidentified in the tetraploid alfalfa reference genome, so it can be considered that more HSF genes can be identified in the tetraploid alfalfa reference genome. In this study, the MsHSF members were identified in the “Xinjiangdaye” autotetraploid genome. The genome position, gene structure, conserved motifs, and cis-acting elements in the promoters of these MsHSF members were determined. The evolutionary relationship and gene replication events of these MsHSF members between alfalfa and Glycine max, M. truncatula, and A. thaliana were comprehensively examined. Moreover, in order to determine their potential roles and the response level of each HSF gene to different stresses, expression patterns of these MsHSF members in six different tissues of alfalfa were analyzed in depth, and their dynamic expression changes under salt stress, drought stress and cold stress were analyzed to preliminarily clarify the functions of different MsHSF genes in response to abiotic stresses in alfalfa. Our results provide valuable information for further clarifying the molecular regulatory mechanisms of MsHSF genes involved in various abiotic stresses in alfalfa in the future.

Point 2: Alkali stress.  This stress is mentioned twice in the article. First, in the introduction (this is appropriate, since we are talking about abiotic factors). The second time was in the caption to figure 8 (inappropriate, since the article is devoted to only three stress factors, and this one is the fourth one).

Response 2: Thanks for this advice. We have made changes in lines 267-269 as follows:

Fig. 8 Expression of MsHSF genes under drought, salt and cold stress conditions. A. Expression of MsHSF genes under drought stress. B. Expression of MsHSF genes under salt stress. C. Expression of MsHSF genes under cold stress. D. Venn diagram of MsHSF genes expressed under the three abiotic stresses.
